# The Impact of Rural Hospital Closures and Health Service Restructuring on Provincial- and Community-Level Patterns of Hospital Admissions in New Brunswick

**DOI:** 10.3390/ijerph19127258

**Published:** 2022-06-14

**Authors:** Dan L. Crouse, Kyle Rogers, Adele Balram, James T. McDonald

**Affiliations:** 1Department of Sociology, University of New Brunswick, Fredericton, NB E3B 5A3, Canada; dcrouse@healtheffects.org; 2New Brunswick Institute for Research, Data and Training, University of New Brunswick, Fredericton, NB E3B 5A3, Canada; krogers4@unb.ca (K.R.); adele.lundy@gnb.ca (A.B.); 3Department of Political Science, University of New Brunswick, Fredericton, NB E3B 5A3, Canada

**Keywords:** hospital closures, geographic access to health care, ambulatory care sensitive conditions

## Abstract

In the early 2000s, the Province of New Brunswick, Canada, undertook health system restructuring, including closing some rural hospitals. We examined whether changes in geographic access to hospitals and primary care were associated with changes in patterns of hospital use. We described three measures of hospital use for ambulatory care sensitive conditions (ACSCs) among adults 75 years and younger annually during the period 2004–2013 overall, and at the community scale. We described spatial and temporal patterns in: age-standardized hospitalization rates, age-standardized incidence of hospital admissions, and rates of admissions via ambulance. Overall, rates and incidence of hospitalizations for ACSCs declined while admissions via ambulance remained largely unchanged. We observed considerable regional variation in rates between communities in 2004. This regional variation decreased over time, with rural areas demonstrating the sharpest declines. Changes in hospital service provision within individual communities had little impact on rates of ACSC admissions. Results were consistent across urban and rural communities and were robust to analyses that included older patients and those admitted for reasons other than ACSCs. Our results suggest that the restructuring and hospital closures did not result in substantial changes to regional patterns or rates of service use.

## 1. Introduction

In response to declining and ageing rural populations, escalating health care costs, and increasing demand for health services in urban centres, most Canadian provinces have rationalized health care services with the triple aim of reducing costs, improving access, and improving outcomes [1,2,3,4]. In some instances, these attempts at rationalization included the closure of hospitals situated in rural areas. The closure of these rural hospitals is a contested and sensitive health policy issue but is often accompanied by a bolstering of primary care services. The impact of these changes on health care services and health outcomes has not been evaluated systematically. Within Canada’s publicly funded and financially constrained healthcare system that guarantees free access to essential physician and hospital services, it remains unclear whether prioritizing primary care over acute care services is the ideal approach.

Generally, there appears to be a scarcity of quantitative research performing secondary analysis of administrative data on this topic in the literature both nationally and internationally. Within the Canadian context, few studies have examined the effect of replacing acute care services with primary care services on health services use, and results in those studies have been inconsistent. For example, a study in Calgary, Alberta found that centralization of coronary artery bypass grafting was associated with increased rates of procedures being performed, shorter hospital stays, and more favourable short-term outcomes [5]. A study in Nova Scotia reported no adverse population-level perinatal consequences associated with reductions in maternity services due to hospital closures [6], and in Saskatchewan, Liu et al. [7] found that hospital closures had no overall negative impacts either on access to inpatient hospital services or to health status.

Conversely, studies in British Columbia have found that an increased distance to hospital was associated with less utilization of hospital-based services generally [8], increased risk of neonatal mortality, and the number of days in neonatal intensive care units [9]. An earlier study in New Brunswick found that distance from the sole tertiary cardiac care facility providing cardiac operations was associated with worse 30-day outcomes after cardiac procedures [10].

As with all Canadian jurisdictions, the New Brunswick’s Medicare system is nearly universal; anyone registered in the provincial or federal Medicare systems is eligible for hospital and primary care services at no personal cost, with the cost falling on either federal or provincial health systems. In the mid-2000s, the province of New Brunswick adopted rationalization strategies such as those seen in Saskatchewan [2] and Nova Scotia [3]. The provincial government implemented a series of health service reforms and restructuring aimed at containing rising expenditures by centralizing acute care services in larger population centres and by replacing rural acute care services with increased primary care services. The conversion of rural acute care hospitals into primary care community health centres (CHC) was a key characteristic of the restructuring. The repurposing of these hospitals varied by facility, but included eliminating acute care beds, emergency services, and surgery services. The transformation of these facilities from hospital to CHC was then completed by increasing primary care services at that location [4]. In 2004, prior to restructuring, there were 30 general/acute facilities in the province, and by 2013, only 22 remained active. Conversely, over the same time period, 10 CHCs were added. Table 1 describes the presence of acute and primary care services for each HCC as of 2013, and Table 2 describes the changes to acute and primary care services over the ten-year period.

In addition to converting rural hospitals to CHCs, the government also established other primary care resources to replace the loss of acute care services. Some of these additions include establishing new CHCs, establishing smaller health centres and increases to extramural services. Similar to the hospital repurposing, the addition of these primary care services was not uniform across the province: Table 3 outlines the changes to acute and primary care services within each HCC over the ten-year period. As such, both urban and rural communities experienced a variety of changes in access, availability, and proximity to primary and acute care services. Further, the removal and addition of primary or acute care services occurred at different times.

In the New Brunswick context, the increases in primary care resources were achieved primarily by reducing (mostly rural) acute care services. This approach was and remains controversial. In response to announced hospital closures in the early aughts, citizens from the various effected communities separately wrote editorials [11], directly engaged with the government of NB [12], launched TV advertisements [13], and joined together to establish a province-wide coalition resisting the changes [14]. Given the highly politicized nature of these decisions, a firm understanding of their wider implications needs to be understood. 

Documenting and understanding how geographical access to different types of care has changed and understanding the effects of those changes has particular importance in New Brunswick: in 2006, the midpoint of the health system reorganization process, 49% of New Brunswickers lived in rural areas, and only three cities (i.e., Fredericton, Saint John, and Moncton) had populations greater than 50,000 [15].

The objective of this study is to examine whether these changes in geographic access to acute vs. primary care resources following the restructuring phase in New Brunswick were associated with changes in patterns of hospital usage at the community and provincial scales. Specifically, we explore acute care admissions associated with ambulatory care sensitive conditions (ACSCs). ACSCs are conditions (e.g., diabetes) for which early and effective primary care can help avoid or decrease rates of hospitalization. ACSC hospitalizations are considered potentially avoidable because they are related to conditions that should not require hospitalizations if managed appropriately through primary care and are often used as an indicator of access to care and of the quality and effectiveness of primary care [16,17,18,19]. In this instance, they represent a useful metric for examining the simultaneous reduction in acute care services and increase in primary care services implemented in different communities in New Brunswick at different times over a ten-year period. 

Overall, we expected that increases in primary care provision would lead to decreases in ACSC hospitalization rates in affected communities, in line with the theory associated with ACSCs. Further, we suspected that the loss of acute care provision in affected communities could have effects on ACSC hospitalizations independent of changes to primary care provision. To test this, we examined spatial and temporal patterns in: age-standardized ACSC hospitalization rates, age-standardized incidence of ACSC hospital admissions, and rates of ACSC admissions via ambulance, during the ten-year period 2004–2013 among individuals under the age of 75. We chose this period, as it includes the period of hospital restructuring as well as several years pre- and post-restructuring that provide a pre-change baseline and allows for more gradual adjustment in health care service access. In addition to our main analysis, we conducted several robustness checks to determine if our results were sensitive to our sample restrictions (ACSCs vs. all hospitalizations, <75 vs. full population). We also undertook a regression analysis of area-level variables on rates of hospitalization that allowed for the inclusion of potentially confounding variables.

## 2. Materials and Methods

All analyses were conducted in the New Brunswick Institute for Research, Data, and Training (NB-IRDT), a research institute at the University of New Brunswick, Fredericton that was established in 2015 through collaboration with several provincial government departments. NB-IRDT provides a central location for researchers to access and link provincial health-related and other administrative datasets, including pseudonymized patient records. Our analyses are based on records from the New Brunswick hospital Discharge Abstracts Database (DAD) linked to the Citizen Database, an analytical file drawn from the Province’s Medicare Registry as well as time and region-specific data on hospital and community health centre infrastructure. We utilized classifications in the Government of New Brunswick’s annual facility expenditure reports to define facilities as hospitals, CHCs, health centres, and coordinating extramural offices. The annual reports were examined across years to determine if changes in classification for specific facilities occurred (capturing the transformation of a hospital into a CHC), if new facilities were listed (capturing the addition of new services in an area), or if facilities were no longer included (capturing the removal of a service). These reports also provided information on hospital bed capacity and changes in that capacity over the sample period. Analyses of the microdata for service use rate generation included only general and acute care facilities as defined by the facility descriptors within the DAD. We excluded from our analyses admissions to dedicated psychiatric and rehabilitation facilities (*n* = 3). Finally, hospitalizations associated with federal health card numbers (e.g., Canadian Armed Forces, Royal Canadian Mounted Police) were not included in the analyses. The pseudonymization process that facilitates access to microdata on the NB-IRDT platform is applied to NB provincial health card numbers; as such, federal health card numbers do not generate valid records during the matching process.

We adopted criteria for defining ACSCs from a recent Statistics Canada Report [19], which included the following conditions: angina; asthma; chronic obstructive pulmonary disorder; congestive heart failure and pulmonary edema; diabetes; grand mal status and other epileptic convulsions; and hypertension and excludes individuals over the age of 75 (see Appendix A for coding used to identify patients). 

We identified three indicators of hospital access which we estimated for each year from 2004 through 2013. 

Age-standardized annual hospitalization rates for ACSCs per 1000 population;Age-standardized annual incidence of hospital admissions for ACSCs per 1000 population;Age-standardized annual rates of hospital admissions for ACSCs via ambulance per 1000 population.

We considered both hospitalization rates and incidence of hospital admissions because comparing their results contributes different information about the intensity of hospital usage—the former measures intensity of hospitalization use which can include multiple visits by the same individuals, while the latter measures the proportion of the population experiencing some episode of hospitalization. 

Following exclusion criteria applied elsewhere, we excluded from our analyses any ACSC hospitalizations for cardiovascular-related interventions [19]. We also excluded all day-surgeries, as in New Brunswick, these are not all reported to a single database, leading to incomplete data. 

We calculated each of these indicators for patients 75 years or younger at the time of admission. They were calculated at the overall provincial and local levels of Health Council Communities (HCCs). The 33 HCCs were defined by the New Brunswick Health Council based on the province’s seven Health Zones and Statistics Canada census subdivisions (i.e., municipal boundaries) for the purpose of combining administrative health service delivery areas with Statistics Canada boundaries for analysis and reporting purposes. All HCCs have a minimum population of 5000 people [20] (Figure 1). 

We also present the results among communities grouped according to peer groups. the New Brunswick Health Council defines peer groups to facilitate within-group comparisons of communities with similar population sizes and broad sociodemographic characteristics. These peer groups generally correspond to categories ranging from relatively urban (A) to relatively rural (D). For example, peer group A is composed of the census subdivisions that comprise the three major cities in the province, namely: Moncton, Saint John, and Fredericton. Individuals were assigned to specific aggregated geographies based on the postal code of residence according to their Medicare public health insurance registration for each year included in the analysis. Individuals admitted to hospitals contribute to the hospitalization rates of their home HCC rather than the hospitalization rates of the HCC where the facility they were admitted to is located. As such, an individual admitted in 2 consecutive years would contribute to the annual rates of their home HCC regardless of whether that HCC experienced a hospital closure.

Each person may contribute multiple records to the numerator in the calculation of the age-standardized annual hospitalization rates and admissions via ambulance, whereas they contribute only a single record to the annual incidence of admission, which describes the proportion of the population that were admitted at least once during the period. We calculated the rates by dividing hospitalizations within an area each year by the corresponding estimates of the population aged younger than 75 years in that same area based on data from the annual provincial citizen registry. HCC rates were then age-standardized to the New Brunswick population for the corresponding year. Indirect age-standardization was chosen because there were concerns about identifying appropriate and consistent ACSC rates for the three indicators that align with the inclusion/exclusion criteria applied in this study.

### Additional Analyses

Hospitalizations for ACSCs may represent an important indicator of timely and/or effective primary care but may not reflect overall patterns of access to hospital care. As such, to test the sensitivity of our results, we replicated all the above analyses using hospitalizations for any reason (excluding those for mental health-related issues, as well as hospitalizations to psychiatric and rehabilitation facilities, as noted above), again, among patients aged younger than 75 years. We also replicated both above analyses among subjects of all ages (i.e., not restricted to those 74 years of age and younger).

In addition, we ran fixed-effect ecological regression models to examine the potential influence of time-varying area-level (i.e., Health Council Community-level) indicators of service provision using SAS base, version 9.4. Specifically, we used PROC GLM to implement three multivariate linear regression models with HCC as the unit of analysis, and the three service use metrics as the dependent variables. For all models, we controlled for the unobserved time-invariant HCC differences using area-specific dummy variables. Independent variables included:Annual count of hospitals in the HCC;Annual count of CHCs in the HCC;Annual count of health centres in the HCC;Annual count of extramural coordinating offices in the HCC;A binary variable describing when and if a hospital was closed or repurposed;A linear time trend variable (takes the value 1 in 2004, 2 in 2005, …, 10 in 2013).

## 3. Results

In 2004, at the start of our study period, the overall age-standardized hospitalization rate for ACSCs was 8.8 per 1000 people; incidence of admissions was 6.8, and admissions via ambulance was 2.4. We present in Figure 2 trends in these overall provincial rates from 2004 through 2013. We observed nearly identical patterns of decline in hospitalization rates and incidence of hospital admissions during the ten-year period (i.e., both decreased by ~54%), while admissions via ambulance remained essentially unchanged during this time.

Neither rates of hospitalizations nor the incidence of hospital admission increased in any Health Council Community during the study period. Tracadie–Shediac experienced the greatest decline in the hospital admissions considered here (i.e., a decrease of 9.2 per 1000 people), whereas Grand Bay–Westfield experienced the smallest (essentially negligible) decline (i.e., 0.3 admissions per 1000). The community of Florenceville–Bristol is the only community that experienced a greater decrease in the incidence of hospital admissions than in the hospitalization rate. Admissions via ambulance did not demonstrate a universal decrease over time that was observed in the other rates. Arrival by ambulance demonstrated little variability, showing neither an increase nor a decrease by two people per 1000 in any community during this ten-year period. Of note is the fact that during the periods when hospitals were repurposed, there were no clear secular changes in hospitalization rates in any of the affected Health Council Communities.

Although the patterns of change were generally consistent across communities, we observed some area-specific differences in overall rates in some years. That is, even though there were few substantial local changes in rates, we observed geographic cross-sectional variability in annual rates. For example, in 2006, the community of Quispamsis had an annual age-adjusted hospitalization rate of 3.2 per 1000, compared to Perth–Andover, which had a rate of 14.1 per 1000 (results not shown). 

The patterns of declining admissions, including timing and magnitude of the decline, were generally consistent across communities of different sizes, although it is notable that the largest declines over the period were for individuals in less populated areas. We present in Figure 3 trends in hospitalizations aggregated by peer group, and although all peer group regions exhibit a clear downward trend, the net result is a convergence in hospitalization rates across regions by the end of the sample period. 

To isolate the effect of hospital closures and other structural changes from broader changes in health service access highlighted above, we present in Table 3 results from the three area fixed effects regression models. The number of CHCs and extramural offices present in an area was associated with decreases in both the rate of hospitalizations and the incidence of hospital admission, with the availability of extramural care seeming to play a critical role. Neither the number of CHCs or the number of extramural offices were associated with the arrival by ambulance rate, although with this metric, the number of health centres in an area became important. 

The provision, or loss of provision, of acute care services through a hospital had smaller, although positive, effects. Generally, the presence of hospitals within a community each year was not associated with either hospitalization rate or incidence of hospital admissions but was associated with slightly reduced admissions via ambulance. Conversely, closure of a hospital led to slight increases in ACSC hospitalizations and incidence of hospital admissions, with little effect on admissions via ambulance. 

We performed several sensitivity analyses to ensure that the data extraction criteria were not influencing our results. As noted above, these included eliminating the age cut-off (<75) and including hospitalizations for all causes (not just those of ACSCs). These analyses returned temporal and geographic trends in all three rates in line with our main findings. Table 4 describes the parameter estimates and statistical significance for three distinct samples: (1) ACSC conditions, no age restriction; (2) all hospitalizations, age-restricted to individuals less than 75 years; and (3) all hospitalizations, no age restriction.

## 4. Discussion

We conducted a population-based analysis of hospital admissions using ten years of individual-level data for all Medicare-eligible residents in New Brunswick, Canada. We examined the impacts of large-scale health care restructuring on the use of health services, restructuring that often replaced acute care service provision with increases in primary care services. This is among only a handful of Canadian studies that have examined such issues. Overall, rates and incidence of hospitalizations for ACSCs declined over the study period, and admissions via ambulance remained largely unchanged. Moreover, changes in hospital service provision within individual communities appeared to have little impact on rates of admissions for these conditions among local populations (i.e., mean change in rates of annual hospitalizations for ACSCs across all communities between 2004 and 2013 was −4.9 admissions per 1000). Specifically, it appears that replacing rural hospitals with CHCs did not have specific effects on hospitalization independent of the broader observed trends of declining hospitalization rates that were in evidence over the full sample period, although given the presence of non-significant positive effects in all regression models, it is possible that the removal of a hospital from an area without subsequent bolstering of primary care services could lead to increases in ACSC hospitalizations. These results were consistent across urban and rural communities and were robust to analyses that included older patients and those admitted for other reasons. Ecological regression analyses supported the interpretation that changes to acute service provision had little effect on hospitalization rates outside of the observed universal decrease over time. Our findings are consistent with those from a much larger American study that also found that the closures of 195 hospitals had no measurable impacts on local rates of hospitalization (or mortality) during a similar period (i.e., 2003–2011) [21].

In addition to the general decrease in ACSC acute care service use across the province, rates across communities appeared to converge towards the provincial average over the ten-year period. As a result, the cross-sectional variation of community-specific rates in 2004 was notably greater than those in 2013. As such, areas with higher rates in 2004 (which were predominantly rural) experienced a greater rate of decline over the ten-year period to produce the observed convergence of rates, but rates of decline across communities did not appear to be affected by hospital closures in some of those communities. We can also infer from the ACSC hospitalizations rate approaching the incidence rate that over time, individuals with ACSCs who go to the hospital are presenting to the hospital fewer times in any given year. Conversely, we noted earlier that Florenceville–Bristol experienced a greater decrease in the ACSC incidence rate than in the hospitalization rate. This finding suggests that relatively few people from this community were seeking acute care for ACSCs, but those seeking care were making multiple trips to the hospital.

The observed trend of decreasing NB hospitalization rates matches decreases in Canada-wide hospitalization rates observed since the mid-1990s, and likely reflects ongoing efforts throughout Canada to shift system usage from acute to ambulatory care services. For example, a CIHI analysis of inpatient and outpatient service usage between 1995 and 2004 demonstrated simultaneous dual trends of increased outpatient rates, and decreases in inpatient rates [22]. Given this general Canadian shift in focus, and the NB government’s specific focus in the 2000s on replacing acute care services with primary care services, it is possible that the observed decrease in hospitalization represents a policy success. Specifically, increased access to primary care services leads to successful management of ACSCs in the community, ultimately resulting in fewer hospitalizations. This possibility was borne out in the regression modelling, where generally, increases in primary care services were associated with decreases in the rates of service use. The number of CHCs, health centres, and extramural coordination offices in an HCC all contributed to decreases in ACSC hospitalization rates.

There are several potential explanations for the large regional cross-sectional variation observed in 2004 that collapsed over time with the area rates eventually clustering around the provincial average in 2013. The first explanation is that the observed trend was one of the intended effects of the health system rationalization. Research has demonstrated that rural and remote Canadians typically have less access to health care services in comparison to their urban counterparts [23,24], and that these urban/ rural differences in health care provision can result in urban/rural health disparities [25]. The observed trend was driven by rural HCCs, and the areas which saw their hospitals repurposed into CHCs were also rural areas. Reducing rural acute care services and putting those resources towards rural primary care services may have allowed rural residents with ACSC to avoid hospitalizations.

A second explanation combines the rural health service disparity with the health of individuals living in rural areas. A CIHI analysis in brief reported that rural areas saw greater rates of ACSCs [26], as well as an increased likelihood of seeking care at emergency departments for conditions that could be handled at the primary care level, concluding that increasing robustness of the rural primary care system could lead to improved health outcomes and decreased acute care service use. Once again, rationalizing the system to decrease rural acute care services, while simultaneously increasing rural primary care services, may have helped to shrink the cross-sectional difference in hospitalization rates between urban and rural areas.

A third potential explanation for the large regional variation observed in 2004 relates to heterogeneous within-province migration patterns. Although the annual age standardization should capture general population shifts, it is possible it was insensitive to health specific population shifts. For example, if 15 individuals living with ACSCs migrated from a small, rural area (e.g., *n* = 400) to a metropolitan area (*n* = 30,000), the underlying populations that serve as the denominator in any hospitalization rate calculation will not change significantly, while the population that contributes to the numerator (all individuals with ACSCs in an area) would. These unequal changes, wherein individuals living with ACSC migrate at a higher rate out of rural areas could contribute to the observed compression effect. Further, it is possible that changes in rural care provision (e.g., a hospital closure) could prompt such intra-provincial migration—individuals with certain conditions could be opting to live closer to urban areas after rural hospital closures.

Two of the explanations for the observed compression of rural hospitalization rates between 2004 and 2013 described above rely on the assumption that at the individual level, an increase in rural primary care services led to an increase in primary care usage, which in turn led to maintenance or improvements of individual health, ultimately leading to the observed decrease in rural ACSC hospitalization rates of the ten-year period. Future research will study those assumptions in greater detail.

This study was limited to describing impacts on health service use that were measurable with available administrative health datasets. For example, we were not able to capture intangible impacts on patients or their families, such as additional costs of transportation or inconveniences related to visiting friends or family due to hospitals being located further from some local communities. As described above, distance from care provision matters, with greater distances leading to decreased service use and effectiveness of service [9,10,17]. Furthermore, we may have found other patterns of change or impacts on admissions if we had considered admissions for specific surgical interventions or those requiring specialist services, including those related to perinatal health (due to loss of local specialists through selective hospital closures). Related to this is the fact that people are not limited to visiting primary care facilities located within their home community, and such simple counts of facilities in a community (as used in our regression analyses) do not completely capture the availability of services to members of that community. Future work could also attempt to describe or control for changes in comorbid conditions of the local populations that are not captured by age, community, and the time trend. 

A key strength of this study is the population-based nature of our datasets, namely that we were able to include all acute care admissions for the outcomes of interest for almost all residents in the province over the ten-year period. We considered three different measures of health service use in our main models based on ACSCs and conducted several sensitivity analyses that considered patients of different ages and those admitted for additional conditions.

## 5. Conclusions

We find that the closure or repurposing of rural hospitals implemented across New Brunswick in the early 2000s had little effect on the ACSC hospitalization rates of those communities who experienced a loss in acute care services, although these may have contributed to the more general trend in decreasing ACSC hospital service usage over time. The reasons for these patterns are likely due in part to the strategy of replacing acute care services with more primary care, and therefore, individuals with ACSCs are being treated successfully through primary care within their communities.

The policy implications of this are clear: placing more primary care providers and resources into the community provides patients with ACSCs support in managing their health conditions. This upstream approach to disease management ultimately helps to reduce the presentation of more severe downstream outcomes that require more intensive care provided by acute care services such as hospitals. For example, a patient living with diabetes who has regular foot-care check-ups from a local clinic runs less of a risk of having complications that result in a hospitalization and possibly an amputation. While preventing the future onset of ACSCs in healthy individuals where possible remains the ideal intervention, increasing primary care resources to help individuals living with ACSCs prevent complications allows the health system to maximize patient quality of life while also increasing long-term sustainability.

Health system rationalization, particularly in rural areas, remains highly controversial. For example, in early 2020, the Government of New Brunswick reversed plans to partially close rural hospital emergency rooms overnight because of public outcry. It is important to emphasize that an increase in primary care resources can be effectively substituted for some acute care services, but policy makers would do well to ensure robust consultations and stakeholder engagement prior to any such changes being implemented.

## Figures and Tables

**Figure 1 ijerph-19-07258-f001:**
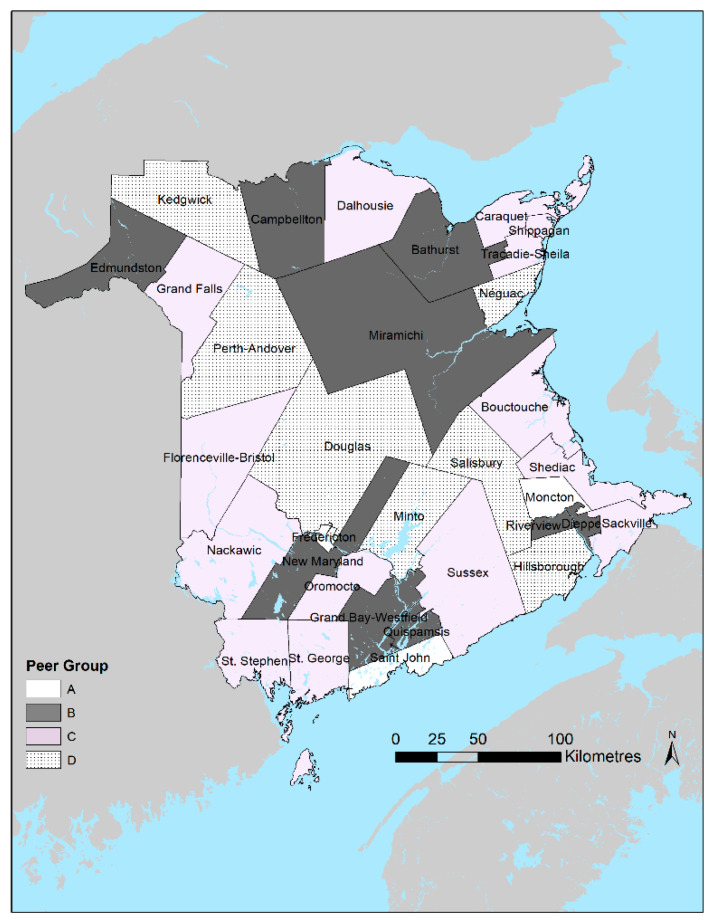
Peer groups for New Brunswick Health Council Communities. Peer Groups A through D represent a continuum where A = Relatively urban, D = relatively rural.

**Figure 2 ijerph-19-07258-f002:**
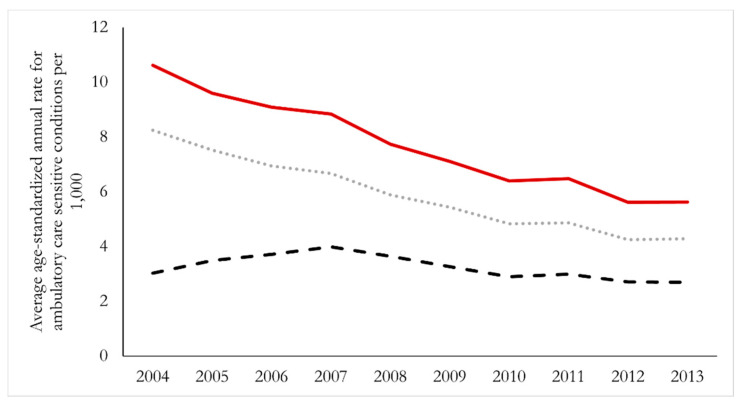
Trends in age-standardized annual rates of hospitalization, incidence of hospitalization, and ambulance arrival for ambulatory care sensitive conditions among those 74 years of age and younger in New Brunswick, 2004–2013. Dash—ambulance arrival rate, Dots—hospitalization incidence, solid line—hospitalizations.

**Figure 3 ijerph-19-07258-f003:**
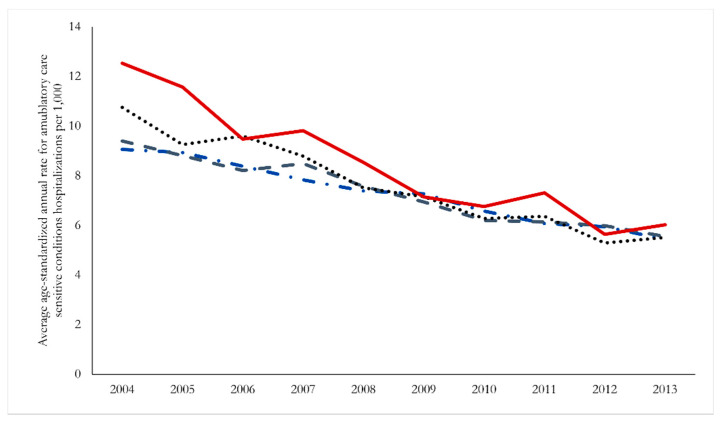
Trends in age-standardized annual rates of hospitalizations for ambulatory care sensitive conditions among those 74 years of age and younger in New Brunswick, 2004–2013 according to community peer groupings. Dash/Dot—major cities, Dashes—smaller urban areas, dots—towns, solid line—rural areas.

**Table 1 ijerph-19-07258-t001:** Provision of acute and primary care services per the Health Council Community (HCC) as of 2013.

Community	Peer Group	Hospitals	Acute Care Beds	Community Health Centre	Health Centre	Extramural Office
Fredericton	A	1	314	2	0	1
Moncton	A	2	684	0	0	1
Saint John	A	2	484	2	0	1
Bathurst	B	1	215	0	1	1
Campbellton	B	1	146	0	0	0
Dieppe	B	0	0	0	0	1
Edmundston	B	1	169	0	0	1
Grand Bay–Westfield	B	1	0	0	0	0
Miramichi	B	1	141	0	3	2
New Maryland	B	0	0	0	0	0
Quispamsis	B	0	0	0	0	1
Riverview	B	0	0	0	0	0
Bouctouche	C	0	0	0	1	1
Caraquet	C	1	12	0	1	1
Dalhousie	C	0	4	1	1	1
Florenceville–Bristol	C	1	52	0	0	1
Grand Falls	C	1	20	0	1	1
Nackawic	C	0	0	0	2	0
Oromocto	C	1	45	1	1	1
Sackville	C	1	21	0	1	1
Shediac	C	0	0	0	1	1
Shippagan	C	1	12	1	1	1
St. George	C	1	8	0	2	1
St. Stephen	C	1	44	0	1	1
Sussex	C	1	25	0	0	1
Tracadie–Sheila	C	1	59	1	0	1
Douglas	D	0	0	1	2	0
Hillsborough	D	0	0	1	0	0
Kedgwick	D	1	6	0	0	1
Minto	D	0	0	1	1	0
Néguac	D	0	0	0	1	1
Perth–Andover	D	0	22	1	0	1
Salisbury	D	0	0	0	1	0

Peer groupings: A—major cities, B—smaller urban areas, C—towns, D—rural areas.

**Table 2 ijerph-19-07258-t002:** Overview of 10-year differences in the provision of acute and primary care services per Health Council Communities: 2004–2013.

Community	Peer Group	Hospitals	Change in Acute Care Beds	Community Health Centres	Health Centres	Extramural Office
Fredericton	A	-	24	↑↑	-	-
Moncton	A	-	21	-	-	-
Saint John	A	-	−43	↑	-	-
Bathurst	B	-	−9	-	-	-
Campbellton	B	-	5	-	-	-
Dieppe	B	-	-	-	-	-
Edmundston	B	-	-	-	-	-
Grand Bay–Westfield	B	-	-	-	-	-
Miramichi	B	-	−32	-	-	-
New Maryland	B	-	-	-	-	-
Quispamsis	B	-	-	-	-	-
Riverview	B	-	-	-	-	-
Bouctouche	C	-	-	-	-	-
Caraquet	C	-	−27	-	-	-
Dalhousie	C	↓	−40	↑	-	-
Florenceville-Bristol	C	↓	−2	-	-	-
Grand Falls	C	-	−15	-	-	-
Nackawic	C	-	-	-	↑↑	-
Oromocto	C	-	-	↑	-	-
Sackville	C	-	-	-	↑	-
Shediac	C	-	-	-	-	-
Shippagan	C	-	-	-	↑	-
St. George	C	-	-	-	↑	-
St. Stephen	C	-	−15	-	-	-
Sussex	C	-	−11	-	-	-
Tracadie–Sheila	C	-	6	↑	-	↑
Douglas	D	-	-	-	-	-
Hillsborough	D	-	-	-	-	-
Kedgwick	D	-	−6	-	-	-
Minto	D	↓	−15	↑	-	-
Néguac	D	-	-	-	-	-
Perth–Andover	D	↓	−26	↑	-	-
Salisbury	D	-	-	-	-	-

Peer groupings: A—major cities, B—smaller urban areas, C—towns, D—rural areas. Each arrow represents a facility.

**Table 3 ijerph-19-07258-t003:** Ecological regression results.

Area-level Variables	Hospitalization Rate			Incidence Rate			Ambulance Arrival Rate		
	Estimate	95% Confidence Limits		Estimate	95% Confidence Limits		Estimate	95% Confidence Limits	
Time	**−0.52**	**−0.57**	−0.47	**−0.42**	−0.46	−0.39	**−0.09**	−0.12	−0.06
Hospital closure	1.34	0.35	2.32	**0.88**	0.18	1.58	0.12	−0.49	0.73
Number of hospitals	0.08	−0.77	0.93	0.02	−0.58	0.63	**−0.76**	−1.28	−0.23
Number of CHCs	**−0.64**	−1.19	−0.09	**−0.48**	−0.88	−0.09	0.03	−0.31	0.37
Number of HCs	−0.60	−1.24	0.05	-0.28	−0.74	0.18	**−0.54**	−0.94	−0.14
Number of extramural offices	**−5.23**	−7.82	−2.64	**−3.72**	−5.56	−1.88	1.50	−0.10	3.10
Number of acute care beds	0.01	−0.01	0.03	0.01	−0.01	0.02	−0.01	−0.02	0.01

CHC—Community health centre, HC—Health Centre. Statistically significant (*p* < 0.05) estimates are **bolded**.

**Table 4 ijerph-19-07258-t004:** Robustness checks.

Area-Level Variables	Hospitalization Rate, Sensitivity Check Parameter Estimates			Incidence Rate, Sensitivity Check Parameter Estimates			Ambulance Arrival Rate, Sensitivity Check Parameter Estimates		
	1	2	3	1	2	3	1	2	3
Time	**−0.75**	**−3.16**	**−3.81**	**−0.57**	**−1.97**	**−2.18**	**−0.17**	**−0.35**	**−0.44**
Hospital closure	**1.51**	3.07	1.74	**0.99**	1.42	0.74	−0.03	−0.04	−1.49
Number of hospitals	−0.05	1.73	1.96	0.03	1.57	1.60	**−1.14**	**−2.76**	**−3.84**
Number of CHCs	−0.71	0.47	−0.04	**−0.55**	0.06	−0.15	0.18	0.51	0.53
Number of HCs	−0.37	−0.84	−1.14	−0.16	−1.27	−1.50	**−0.60**	−0.62	−1.50
Number of extramural offices	**−5.91**	**−16.65**	**−31.19**	**−3.90**	**−10.65**	**−13.63**	**2.88**	3.37	2.96
Number of Acute care beds	0.02	−0.01	0.06	0.01	0.00	0.03	−0.01	−0.05	−0.06

CHC—Community health centre, HC—Health Centre. Statistically significant (*p* < 0.05) estimates are **bolded**; Sensitivity Check 1—ACSC, all ages.

## Data Availability

The datasets generated and analyzed during the current study are not publicly available due to patient confidentiality but are available for access to researchers with approved access to the secure facilities of the New Brunswick Institute for Research, Data, and Training: https://www.unb.ca/nbirdt (accessed on 7 June 2022).

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
