# Peer review of "The Impact of Rural Hospital Closures and Health Service Restructuring on Provincial- and Community-Level Patterns of Hospital Admissions in New Brunswick"

_ijerph, 2022, doi:10.3390/ijerph19127258_

Round 1

Reviewer 1 Report

Dear authors , the paper topic is interesting and relevant. Some clarifications are needed to improve readability and clarity

-       In the introduction a summary/table on other countries examples can be included with highlight of good practices/strategies to be implemented

-       Clarification of study objective at the end of introduction section can improve readability 

-       Figure 2 and 3 lack of a legend into the image

-       Limitations and future developments should be highlighted in the conclusion section

-       Implications for research and impacts for policy makers, healthcare planning and Public Health must be considered in the discussion/conclusion section

Author Response

Dear authors, the paper topic is interesting and relevant. Some clarifications are needed to improve readability and clarity:

In the introduction a summary/table on other countries examples can be included with highlight of good practices/strategies to be implemented

We found little evidence of additional comparable, quantitative analyses beyond the few examples that we mention already in the Introduction.  We have, therefore, indicated further that there is a scarcity of research on this topic.

 Clarification of study objective at the end of introduction section can improve readability

We have revised the text in the Introduction to include the objectives of the study: “The objective of this study is to examine whether these changes in geographic access to acute vs. primary care resources following the restructuring phase in New Brunswick were associated with changes in patterns of hospital usage at the community and provincial scales. Specifically, we explore acute care admissions associated with ambulatory care sensitive conditions (ACSCs).”

 Figure 2 and 3 lack of a legend into the image

Please note that the Figure captions indicate what each line in each figure refers to (e.g., for Figure 2, the caption indicates the following: “Dash = ambulance arrival rate; Dots = hospitalization incidence; solid line = hospitalizations.”

 Limitations and future developments should be highlighted in the conclusion section

These are addressed beginning at line 392.  Limitations that we discuss include: a) the fact that we were only able to describe impacts on health services that are measurable with administrative datasets (i.e., we weren’t able to evaluate inconveniences/challenges related to travelling greater distances to receive care); b) we did not consider health outcomes that required surgical interventions or specialist services; c) we also acknowledge that people may not always choose to visit the health provided that is located closest to where they live.
We note also that future analyses could attempt to describe or control for changes in comorbid conditions of the local populations that are not captured by age, community, and the time trend (as were captured here).

 Implications for research and impacts for policy makers, healthcare planning and Public Health must be considered in the discussion/conclusion section

We have added a new paragraph describing these implications in the Conclusion (beginning at line 420).

Reviewer 2 Report

This paper sets out to report changes in geographic access between acute (hospital) and primary care (ambulatory) uses following the restructuring of these organisations in Canada. The authors conclude that that the restructuring and hospital closures did not result in substantial changes to regional patterns or rates of service use. 

Overall the paper is well written and covers an area of importance for policy makes globally. Two issues that the authors may wish to add to help other systems considering such changes include:

(a) How the public viewed these changes and politically how this was handled, and

(b) Whether the changes meant that for certain specialties staff worked in primary care on outreach programmes from the hospitals or whether the service configuration meant that the staff within the hospitals were redeployed elsewhere.

Overall this is a valuable piece of work. 

Author Response

Reviewer 2

This paper sets out to report changes in geographic access between acute (hospital) and primary care (ambulatory) uses following the restructuring of these organisations in Canada. The authors conclude that that the restructuring and hospital closures did not result in substantial changes to regional patterns or rates of service use. 

Overall the paper is well written and covers an area of importance for policy makes globally. Two issues that the authors may wish to add to help other systems considering such changes include:

  • How the public viewed these changes and politically how this was handled, and

We have added new text in the Introduction to explain this context (Lines 90-97) and note that this is an ongoing issue in the Conclusion (beginning at line 432).

  • Whether the changes meant that for certain specialties staff worked in primary care on outreach programmes from the hospitals or whether the service configuration meant that the staff within the hospitals were redeployed elsewhere.

Data on facility personnel are not available to us, but the amount of redeployment would likely vary depending on the context of each individual community/closure.  We would expect there to be redeployment of a significant proportion of staff given the rural nature of the communities affected.

Overall this is a valuable piece of work. 

We thank this reviewer for this comment.

Reviewer 3 Report

General remarks

The paper deals with a very interesting and timely topic— it examines the impacts of large-scale health care restructuring on the use of health services. More of such studies are needed in the Canadian context. The paper is comprehensively referenced to existing literature, and the methodology is aligned with the purpose of the study. That said, a few things need tightening though to further enhance the quality of the paper.

Conclusion

-The conclusion is too brief, so does not do justice to the richness of the study. More precisely, the authors should mention the implications of their findings both in terms of policy and future research.

Author Response

Reviewer 3

General remarks

The paper deals with a very interesting and timely topic— it examines the impacts of large-scale health care restructuring on the use of health services. More of such studies are needed in the Canadian context. The paper is comprehensively referenced to existing literature, and the methodology is aligned with the purpose of the study. That said, a few things need tightening though to further enhance the quality of the paper.

 Conclusion

-The conclusion is too brief, so does not do justice to the richness of the study. More precisely, the authors should mention the implications of their findings both in terms of policy and future research.

We thank this reviewer this feedback.  We have now expanded on Conclusion substantially.
